# Octacalcium Phosphate for Bone Tissue Engineering: Synthesis, Modification, and In Vitro Biocompatibility Assessment

**DOI:** 10.3390/ijms222312747

**Published:** 2021-11-25

**Authors:** Anastasia Yu. Teterina, Igor V. Smirnov, Irina S. Fadeeva, Roman S. Fadeev, Polina V. Smirnova, Vladislav V. Minaychev, Margarita I. Kobyakova, Aleksandr Yu. Fedotov, Sergey M. Barinov, Vladimir S. Komlev

**Affiliations:** 1A.A. Baikov Institute of Metallurgy and Material Science Russian Academy of Sciences, Leninskiy Prospect 49, 117334 Moscow, Russia; baldyriz@gmail.com (I.V.S.); mikheevap7@gmail.com (P.V.S.); vminaychev@gmail.com (V.V.M.); kobyakovami@gmail.com (M.I.K.); fedotov.ayu@gmail.com (A.Y.F.); barinov_s@mail.ru (S.M.B.); 2Institute of Theoretical and Experimental Biophysics, Russian Academy of Sciences, 142290 Pushchino, Russia; fadeevrs@gmail.com

**Keywords:** biocompatible materials, calcium phosphate compounds, hydroxyapatite, octacalcium phosphate, bone tissue regeneration, biomineralization

## Abstract

Octacalcium phosphate (OCP, Ca_8_H_2_(PO_4_)_6_·5H_2_O) is known to be a possible precursor of biological hydroxyapatite formation of organic bone tissue. OCP has higher biocompatibility and osseointegration rate compared to other calcium phosphates. In this work, the synthesis of low-temperature calcium phosphate compounds and substituted forms of those at physiological temperatures is shown. Strontium is used to improve bioactive properties of the material. Strontium was inserted into the OCP structure by ionic substitution in solutions. The processes of phase formation of low-temperature OCP with theoretical substitution of strontium for calcium up to 50 at.% in conditions close to physiological, i.e., temperature 35–37 °C and normal pressure, were described. The effect of strontium substitution range on changes in the crystal lattice of materials, the microstructural features, surface morphology and biological properties in vitro has been established. The results of the study indicate the effectiveness of using strontium in OCP for improving biocompatibility of OCP based composite materials intended for bone repair.

## 1. Introduction

The bones regeneration processes studies, rather than mechanical replacement of bone tissue, lead to the creation of various combinations of materials based on calcium phosphate ceramics and its composites for bone augmentation procedures [1,2,3]. A large number of works are devoted to bioresorbable materials for maximum imitation of the structure and properties of native bone tissue [4]. This led to the development of matrices and scaffolds with specified physicochemical properties based on biocompatible calcium phosphates—hydroxyapatite and tricalcium phosphate [5,6,7]. The presence of calcium and phosphorus ions in the composition of these materials imparts osteoconductive properties to the implant. The presence of calcium and its release from the material during the healing process can mediate the expression of osteopontin, which binds to bone hydroxyapatite, participates in its formation, and has an integrin-binding domain, which promotes cell proliferation on the implant surface [8]. It is well-known that calcium phosphate materials obtained by high-temperature routes are biocompatible, non-toxic, but devoid of the possibility of biointegration, have a low rate of resorption and can cause fibrous encapsulation of the entire material at the site of implantation [9,10,11,12]. Therefore, a promising approach for dental or orthopedic reconstructive surgery is using the low-temperature calcium phosphate ceramics, which are the most appropriate as possible to the components of the native bones, in particular apatite-like hydroxyapatite (HAp) and its possible precursors—dicalcium phosphate dihydrate (DCPD) and octacalcium phosphate (OCP) [13]. It was previously described that these materials, synthesized under conditions as close as to natural biominerelization processes, can have a direct inducing effect [14]. First, such materials include OCP, which, unlike other analogs, has a high resorption rate and pronounced osteoinductive potencies. The maim interest in materials for bone tissue regeneration based on OCP is also associated with its structural characteristics, which allow the introduction of cations and anions, as well as biological agents for targeted functionalization [15,16,17]. There is no unambiguous opinion about the role of ionic substitutions in the physiology of bone tissue, but it has been revealed that it is possible to change and improve the properties of materials due to structural substitutions with cations and anions. At the same time, depending on the nature and concentration of the embed ion, both improvement and deterioration of the functional properties of the implant is possible. Thus, it is possible to obtain materials based on calcium phosphates with both high bioactivity and antibacterial properties [18,19]. Since the low-temperature OCP considered in this work has a suitable bioresorption rate, the currently unsolved problem is to increase the biological properties of the obtained material. Its osteoinductive properties are important, which are proposed to be improved by substitution of calcium for strontium. The process of reparative osteoregeneration begins with the intensification of the proliferative activity of osteoblasts [20,21]. Strontium-containing materials optimize osseointegration on the initial stages, shifting the balance between osteoplastic synthesis and osteoclastic resorption. Strontium ions in calcium phosphate contribute to the improvement of osteoblast function and subsequent bone formation [22].

That is the reason why, the main goal of this work was to develop an approach for creating a low-temperature OCP, a possible precursor of biological hydroxyapatite, with strontium ions included into the structure. In addition to assessing the effect of the concentration of calcium substitution to strontium on phase and structural changes in the material, an important objective was to assess the effect on biological characteristics.

## 2. Results

XRD patterns of obtained OCP and OCP-Sr are shown in Figure 1. It was found that the samples are solid solutions based on octacalcium phosphate pentahydrate (Ca_8_H_2_(PO_4_)_6_·5H_2_O) and correspond to card № 26-1056 of the XRD base ICDD (Powder Diffraction File, Alphabetical Index Inorganic Compounds, Pensylvania: JCPDS, 1997). With an increase in the percentage of strontium replacing calcium, the main diffraction peaks (about 4.9° and 26.4°) shift to the left relative to the initial OCP. In this case, the shear angle also enlarges with an increase in the strontium concentration in OCP. Diffraction peaks for all examined samples are widened and overlapping peaks are observed at far corners, which are accompanied by an increase in the volume of unit cells. The values of the parameters of triclinic unit cells (Table 1) were set using the CelRef program for eight reflections ((020), (110), (-101), (-211), (002), (260), (-1-42), and (070)). The findings indicate that the parameters of the crystal lattice gradually increase with an increase in the strontium concentration. A higher degree of crystallinity is observed in samples OCP-Sr_5, OCP-Sr_10, and OCP-Sr_20. These data are well consistent with the results of E. Boanini, M. Gazzano, and A. Bigi (2010) [23], where the values of 1226 Å3 and 1231 Å3 are given for similar samples with Sr content of 10 and 15 at.%, respectively. The crystallinity of the samples decreases with an increase in strontium concentration and leads to the formation of less stable materials with lower crystallinity. The inclusion of strontium with a larger ionic radius (0.12 nm) than calcium (0.10 nm) leads to broadening of the crystal lattice parameters and causes relaxation of the lattice of surrounding atoms and can stabilize the metastable structure of OCP crystals and promote the formation of apatite in native bone tissue during regeneration [24]. The diffraction peaks of HA and DCPD were not detected, therefore, the obtained samples are single-phase and correspond to OCP.

The data obtained by the method of IR spectroscopy confirm the data of the X-ray phase analysis. FTIR spectrum presented in Figure 2. The sharp shapes of bands indicate high crystallinity. The presence of the intense bands at 1123 and 1027 cm^−1^ corresponding to the ν3 mode of HPO_4_^2−^ and PO_4_^3−^ [P−O stretching in phosphate (PO_4_^3−^) and hydrogen phosphate (HPO_4_^2−^)], and a HPO_4_^2−^ band at 906 cm^−1^ (P−O stretching in HPO_4_^2−^) are typical of an OCP structure. The 602 cm^−1^ and 560 cm^−1^ sharp P−O bands (P−O deformation in PO_4_^3−^), which is the ν4 mode of PO_4_^3−^. The bands at 865 cm^−1^ and 960 cm^−1^ belong to HPO_4_^2−^ stretch groups (P−OH). These main vibration peaks of PO_4_^3−^ and HPO_4_^2−^ groups are typical of the OCP structure.

The structure and surface morphology of the OCP and OCP-Sr samples were observed using SEM, the results are shown in Figure 3. The intrinsic OCP lamellar crystal shape is observed in all samples. The crystals are enlarged with an increase in samples’ strontium concentration, and the shape approaches the plate. The micrographs revealed the most crystallized structure of the samples (Figure 3c,d). With an increase in the strontium content, the morphology becomes uneven; in the case of OCP-Sr_50, the crystals have a wide lamellar shape, rough edges, and an advanced surface. At the minimum strontium content (1 at.%, Figure 3b), the shape of the crystals is close in its macrostructure to the DCPD, from which the material was synthesized. Analysis of the microstructure of samples OCP-Sr_10 and OCP-Sr_20 showed two types of particle morphology, in addition to lamellar particles, some of the particles have a complex shape, which indicates inhomogeneity. The OCP-Sr_10 samples show structures consisting of small crystals of complex shape. The data of diffraction methods, including EDS analysis and XRD, did not confirm the presence of other phases in the obtained samples. According to the results of diffraction research methods, it was proved that the inclusion of strontium in the OCP structure during low-temperature synthesis occurs in a wide range of substitution of Sr for Ca and changes the integrity of OCP crystals.

The chemical composition of materials was determined using EDA and X-ray fluorescence analysis (Table 2). The results obtained using two different methods are well concorded and confirm the earlier conclusions. OCP fits in the structure of the material in a wide range of concentrations. The strontium concentrations obtained during the experiment are comparable with the theoretical ones, except for the substitution of OCP–Sr_50. An overestimated strontium content can presumably destroy the OCP structure, as shown by previous methods. The molar ratio Ca + Sr\P in the OCP–Sr_5 and OCP–Sr_10 samples is close enough to the theoretical value of 1.33. Thus, strontium can improve the stability of OCP within certain concentrations, above which it has a negative effect on the crystallinity of the structure.

The data of the analysis of the OCP-Sr samples obtained by the TEM method are presented in Figure 4. The instrument constant for all SAEDs is 25.1 mm A. The red circles indicate the diffraction area. Two structural types of synthesized samples are considered—compact, and with needle-shape crystals with directional growth on the surface. Particles of clear OCP look like homogeneous single crystals (Figure 4A) and 20–30% particles, as in Figure 4B. Most of the OCP-Sr_10 particles look like homogeneous single crystals Figure 4C. About 5–10% of particles in the powder have a complex structure (Figure 4D). The contrast within particles indicates possible heterogeneity in samples. The character of the selected areas electron diffraction patterns indicates a microcrystalline structure. Bent extinction contours point to a stressed state.

As a result, the synthesized material for the study was a powder of OCP and OCP-Sr with a highly developed surface due to the radial arrangement of thin crystals.

The effect of strontium-substituted OCP variants on the viability of C_3_H/10T1/2 cells was investigated. Cell viability after incubation with OCP with 1% calcium substitution for strontium (OCP-Sr_1) was not shown difference from cells incubated with the original OCP (without substitution of calcium for strontium). Cell viability after incubation with OCP with 5% and 10% calcium substitution for strontium (OCP-Sr_5 and OCP-Sr_10, respectively) was insignificantly higher than the cell viability after incubation with native OCP. On the other hand, cell viability after incubation with OCP with 20% and 50% calcium substitution for strontium (OCP-Sr_20 and OCP-Sr_50, respectively) was significantly higher than cell viability after incubation with OCP (Figure 5). Substitution of calcium for strontium in OCP not less than for 20% theoretical can significantly reduce the cytotoxicity of OCP. In further experiments to investigate the effect of OCP and OCP-Sr on C3H/10T1/2 cells, the concentration with minimal effect on cell viability (1 mg/mL) was used for OCP-Sr_20 and OCP-Sr_50. At a given concentration, there is a maximum discrepancy in the effect on cell viability for OCP-Sr and OCP.

It is clear that changes in the mitochondrial membrane potential (ΔΨm) and the content of acidic compartments, such as lysosomes, caused by various inducement are one of the key events of cell damage is shown [25,26].

Thereby, the mitochondrial membrane potential and the content of acidic compartments in cells after 96 h of co-incubation with OCP and OCP-Sr were investigated. LysoTracker Green was used to estimate the content of acidic compartments, mainly lysosomes [27], and to assess the mitochondrial membrane potential (ΔΨm) DIOC_3_(6) [28], followed by flow cytometry. When studying the effect of OCP and OCP-Sr on the mitochondrial membrane potential in C_3_H/10T1/2 cells, it was shown that the incubation of cells with both OCP and all various OCP-Sr, regardless of the degree of substitution of calcium for strontium, reduced ΔΨm in cells, after 96 h of co-incubation, the difference from the untreated control is significant, *p* < 0.05 (Figure 6a). Further, the study of the content of acidic compartments after co-incubation of cells with OCP and OCP-Sr for 96 h showed that incubation with OCP increased the content of acidic compartments in C_3_H/10T1/2 cells, the difference from the untreated control was significant, *p* < 0.05 (Figure 6b). On the other hand, incubation of cells with all OCP-Sr for 96 h did not lead to an increase in the content of acidic compartments in the cells (Figure 6b).

Additionally, to verify flow cytometry data, the content of acidic compartments and mitochondrial membrane potential in C_3_H/10T1/2 cells were examined using confocal microscopy after 96 h of co-incubation with OCP and OCP-Sr. For this, cells were stained with LysoTracker Green and MitoTracker Red CMXRos [29]. H33342 was used to visualize the cell nucleus. Confocal microscopy data were similar to flow cytometry data and showed an increase in the content of acidic compartments in C_3_H/10T1/2 cells after 96 h of co-incubation with OCP, but not with OCP-Sr. Also, for OCP and for all OCP-Sr variants ΔΨm decreased (Figure 7).

It is well known that reactive oxygen species (ROS) play a key role both in the regulation of normal cell physiology and in various cellular pathologies, including cell damage [30]. In this regard, the effect of OCP and OCP-Sr on ROS production in C_3_H/10T1/2 cells was investigated. Co-incubation of cells with OCP, with OCP-Sr_1 and with OCP-Sr_10 for 96 h did not lead to a change in ROS production. However, after incubation of cells for 96 h with OCP with a maximum strontium substitution of theoretical 50 at.% (OCP-Sr_50), a decrease in ROS production was shown, the difference from the control was significant, *p* < 0.05 (Figure 8). It is due to the fact that Sr^2+^ has an antioxidant effect on mesenchymal cells and, probably, this concentration of Sr in the OCP composition may be sufficient to reduce the production of ROS.

The results of in vitro studies show that substitution of calcium for strontium, from 20% substitution and above, in the composition of OCP can significantly reduce the cytotoxicity of OCP. However, regardless of the substitution degree, OCP-Sr, as well as OCP, reduce the mitochondrial membrane potential, but this effect does not correlate with the effect of OCP and OCP-Sr on cell viability. In addition, OCP-Sr does not increase the content of acidic compartments, unlike OCP, however, this effect also does not correlate with the effect of OCP and OCP-Sr on cell viability. The effect of OCP and OCP-Sr on ROS production in cells shows that most of the studied OCP-Sr variants, as well as OCP did not change constitutive ROS production; however, OCP-Sr with a 50% substitution degree of calcium for strontium significantly reduced ROS production in cells C_3_H/10T1/2.

## 3. Discussion

The development of osteoplastic materials based on synthetic calcium–phosphate compounds, equal in efficiency to autografts of bone tissue, is one of the most demanded tasks of modern tissue engineering. However, direct or indirect cytotoxicity, limited biocompatibility, and inability for full biointegration in the recipient’s body are the main limiting factors for the widespread use of calcium phosphate ceramics in current clinical practice [31,32]. In this regard, the transition to the synthesis of calcium phosphate compounds under physiological conditions and the development of biomimetic materials using bioactive synthetic analogs of the natural bone tissue biomineralization process—primarily OCP—open up new prospects for the development and application of osteoplastic materials with high regenerative potential [33].

In this study, strontium ions were doped into the OCP structure by chemical recrystallization of DCPD powder in solution under physiological conditions. The molar ratio of strontium in the samples is close to the theoretical in all samples, except for OCP-Sr_ 50. The substitution of strontium in this case was experimentally up to 25–27 at.%, which is higher than in existing studies. Strontium substitution leads to the expansion of the crystal lattice of OCP and at 5–10 at.% possibly stabilizes the crystal lattice, which requires a more detailed study of the processes occurring in this range.

The results of in vitro biocompatibility studies show that substitution of strontium in samples with OCP-Sr_20 and higher significantly increases cells viability, compared to undoped OCP. The results of in vitro biocompatibility studies show that substitution of strontium in samples with OCP-Sr_20 and higher significantly increases cells viability, compared to undoped OCP. Similar results were obtained in the work of Shi et al., 2017, where it was shown that an increase in strontium incorporation into OCP obtained by the high-temperature method can significantly increase the viability and proliferative activity of mesenchymal stem cells, compared to undoped OCP [34]. This effect may be associated with the absence of the OCP-Sr impact on the content of acidic compartments, as well as with the lack or even an inhibitory effect on the ROS production. The antioxidant properties of strontium in the composition of calcium phosphate compounds have been well studied. In the study by Zhou et al., 2019, it was shown that the use of Sr substituted OCP can reduce the production of ROS in mesenchymal stem cells [35]. The mechanisms of the antioxidant action of strontium-substituted calcium phosphate compounds may be associated with an increase in the activity of antioxidant enzymes, such as superoxide dismutase (SOD), catalase (CAT), and glutathione peroxidase [36]. It has also been shown that Sr2 + ions are able to inhibit lipid peroxidation and, thus, reduce cellular damage caused by oxidative stress [37]. ROS play an important role in the differentiation of bone cells, primarily osteoclasts, responsible for bone resorption [38]. Strontium inhibits osteoclast formation and promotes osteoblast differentiation [39]. Thus, it can be assumed that the antioxidant properties of Sr^2+^ may be related to its osteogenic properties.

The effect of strontium-containing calcium phosphate compounds on the content of acidic compartments, primarily lysosomes, in cells requires further research. However, it is known that the cytotoxic effect of calcium phosphates can be associated with the rapid dissolution of these compounds in lysosomes with their subsequent rupture and cell necrosis [40]. It is also known that endosomes damage in cells after absorption of calcium phosphates can be retired by calcium chelators [41]. Probably, strontium containing calcium phosphate compounds can degrade more slowly in phagolysosomes of cells than calcium phosphate compounds without strontium and, therefore, do not induce damage and biogenesis of lysosomal acidic compartments in cells. However, the exact mechanisms of the effect of calcium phosphate compounds containing strontium on the Red/Ox status of cells and lysosomal physiology remain unclear.

As a conclusion, the inclusion of strontium substituted OCP (OCP-Sr_20 and higher) in composite osteoplastic materials could be significantly increases the efficiency of bone augmentation procedures in dental and orthopedic surgery, as well as reconstructive procedures for correction of post-traumatic defects. It is well known from the literature and clinical practice that strontium preparations successfully suppress bone resorption in the treatment of osteoarthritis and osteoporosis. [42,43], and strontium-substituted calcium phosphate compounds to modify bone balance towards the osteogenic pathway, can enhance the proliferation of preosteoblasts, reduce the production of pro-inflammatory cytokines and matrix metalloproteinases, and suppress the production of chemokines-attractants of immune cells [44,45,46,47], initiating the rejection of the implanted material.

Based on the in vitro data obtained, OCP-Sr (20 and higher) has a higher biocompatibility and can inhibit the biogenesis of lysosomes, which is of great clinical importance, since it can help prevent the side effect of resorption of contact healthy bone tissue during implantation of calcium phosphate materials. A potential increase in the biointegration of OCP-Sr materials can be realized both by local suppression of ROS production by strontium, and by a decrease in the total number of lysosomes and a potential suppression of the production of matrix metalloproteinases in immune cells, which will be clarified at further stages of our research.

## 4. Materials and Methods

### 4.1. Synthesis Procedure

OCP and strontium-doped OCP (OCP-Sr) samples were prepared by low-temperature chemical crystallization of DCPD in an acetate aqueous solution. Powder DCPD with a particle size of 50 µm–100 µm was synthesized by precipitation method. The solutions of (NH_4_)_2_HPO_4_ 0.5 M and Ca(NO_3_)_2_ 0.5 M (Sigma-Aldrich, St. Louis, MO, USA) were mixed together in a glass cup and left for 2 h stirring, after air-drying for 4 days. OCP was synthesized by low-temperature chemical transformation method in 1.75 M sodium acetate solution from DCPD powder by following reaction:$$8{\mathrm{CaHPO}}_{4}\cdot 2H_{2}O={\mathrm{Ca}}_{8}{\left({\mathrm{HPO}}_{4}\right)}_{2}{\left({\mathrm{PO}}_{4}\right)}_{4}\cdot 5H_{2}O+2{\mathrm{HPO}}_{4}^{2-}+4H^{+}+11H_{2}O$$

DCPD powder mass to solution volume ratio was 1g to 100 mL. The process was conducted on a magnetic stirrer at 250 rpm and heating at 35 °C for 24 h. After that, filtration and five-fold washing with distilled water were accomplished. OCP-Sr powder was synthesized by the same method but in presence of Sr^2+^ ions. Strontium nitrate was used to obtain OCP-Sr. The required amount of strontium cations was calculated based on the theoretical substitution of calcium cations in the range from 1 to 50 at.% according to the reaction:$$\left(8-x\right){\mathrm{CaHPO}}_{4}\cdot 2H_{2}O+\mathrm{xSR}={\mathrm{Ca}}_{8-x}{\mathrm{Sr}}_{x}{\left({\mathrm{HPO}}_{4}\right)}_{2}({\mathrm{PO}}_{4}){}_{4}\cdot 5H_{2}O+\;\left(2-x\right){\mathrm{HPO}}_{4}^{2-}+4H^{+}+\left(11-2x\right)H_{2}O$$

For x ≤ 2 strontium substitution is equal to 25 at.% of the theoretical amount. All chemicals were analytical reagents (purity > 99%) Scharlau (Spain). The marking of the samples corresponds to the concentration of strontium introduced into the solution during the preparation of OCP powder, according to the calculated data based on the given reactions. In the study, the OCP-Sr sample corresponds to the material, without the inclusion of strontium in the structure, OCP-Sr_5—the inclusion of strontium 5 at.%, OCP-Sr_10—inclusion of strontium 10 at.%, OCP-Sr_20 and OCP-Sr_50—20 and 50 at.%, respectively.

### 4.2. Characterization

Phase components of powder samples were identified by X-ray diffraction on an Ultima IV diffractometer from Rigaku (Tokyo, Japan) (CuKα radiation, nickel filter, tube voltage 40 kV, tube current 30 mA, high-speed detector D/teX, angular range 4.0°–40.0°, detector movement speed 0.5°/min, step 0.02°). The lattice cell parameters were calculated by the least squares method using the PDXL software package from Rigaku (Tokyo, Japan) using the PDF2 diffractometric data base. The values of the parameters of triclinic unit cells were set using the CelRef program for eight reflections ((020), (110), (-101), (-211), (002), (260), (-1-42) и (070)). The IR-spectra were recorded by Nikolet Avatar 330 spectrophotometer (USA) in the 4000–400 cm^−1^ wavelength region. The KBr pellet technique was used with 1 mg of powder in 50 mg of spectroscopic-grade KBr.

Surface morphology was examined by SEM Tescan VEGA II (Brno, Czech), equipped with energy dispersive spectroscopy systems (EDS; INCA Energy Oxford Instruments, Abingdon, UK), and samples previously were covered with gold by Q150R Quorum Technologies (Lewes, UK). The substructure of the samples was determined by high resolution transmission electron microscopy (TEM) (Philips EM-430 ST device).

Qualitative and quantitative analysis of the samples elemental composition was carried out on a sequential type X-ray fluorescence wave-dispersive spectrometer BRUKER S8 Tiger (series 2) in helium using a standard-free technique using the QUANT-EXPRESS software (Berlin, Germany).

### 4.3. Cell Culture

Murine embryonic mesenchymal cell line C_3_H/10T1/2 was obtained from ATCC (Wesel, Germany). Cells were grown in Basal Medium Eagle (Sigma-Aldrich, Milwaukee, WI, USA) supplemented with heat-inactivated fetal bovine serum (Gibco, Waltham, MA, USA) to a final concentration of 10% and 2 mM L-glutamine (Sigma-Aldrich, St. Louis, MO, USA), 40 μg/mL gentamicin sulfate (Sigma-Aldrich, St. Louis, MO, USA), under conditions of 5% CO_2_ content in the air and at 37 °C. Cells of 7–10 passages were used in the experiments. The cells were planted in 96-well plates (SPL life science, Pocheon, Korea) at a concentration of 1.5 × 10^4^ cells/cm^2^, cultured for 24 h and then added to OCP and OCP-Sr at concentrations of 10.00, 3.33, 1.11, 0.37, 0.12, 0.04, and 0.01 mg/mL and were co-cultured for another 96 h. OCP and OCP-Sr samples were pre-sterilized with 75% ethanol according to the indicated method [48].

### 4.4. Cell Viability Assay

Cell viability after incubation with OCP and OCP-Sr was evaluated by AlamarBlue (Invitrogen, Carlsbad, CA, USA). 100 μg/mL of AlamarBlue was added to the cells after 96 h of incubation. The cells were then incubated for 4 h at 37 °C and 5% CO_2_ content in the air, then the fluorescence intensity was measured at an excitation wavelength of 560 nm and an emission wavelength of 595 nm using an Infinity F 200 plate reader (Tecan, Männedorf, Switzerland). Cell viability was assessed by the mean fluorescence intensity (MFI) of the resulting resofurin product. The viability of control cells not incubated with OCP and Sr-OCP was taken as 100%. Cell viability after incubation with OCP or Sr-OCP was calculated as a percentage relative to control by the formula: Cell viability% = (MFI cells after incubation with OCP and Sr-OCP/MFI control cells) × 100%. Evaluation of the effect of OCP and OCP-Sr on cells was conducted using a trypan blue exclusion assay [49].

### 4.5. LysoTracker Staining

To assess the acidic compartments in the cells after 96 h of co-incubation with OCP and OCP-Sr, the cells were washed three times with phosphate-buffered saline (PBS) (Sigma-Aldrich, St. Louis, MO, USA), then detached from the plastic surface using 0.05% trypsin-EDTA solution (Gibco, Waltham, MA, USA) and stained with 50 nM LysoTracker Green DND-26 (Thermo Fisher Scientific, Waltham, MA, USA) for 30 min in a CO_2_ incubator. Control cells were incubated with 50 μM chloroquine (Sigma-Aldrich, St. Louis, MO, USA) for 4 h. The measurement was conducted using a BD Accuri C6 flow cytometer (BD Sciences, Franklin Lakes, NJ, USA). A total of 3 × 10^4^ cells were analyzed for each sample.

### 4.6. Measurement of Mitochondrial Membrane Potential

The cells were washed three times with PBS after 96 h of co-incubation with OCP and OCP-Sr, then detached from the surface of the culture plastic using 0.05% trypsin-EDTA solution and stained with 10 nM 3,3′-dihexyloxacarbocyanine iodide (DiOC6 (3)) (Sigma-Aldrich, St. Louis, MO, USA) for 30 min in a CO_2_ incubator to measure mitochondrial membrane potential (ΔΨm). As a control, cells were incubated with 250 nM valinomycin (Sigma-Aldrich, St. Louis, MO, USA) for 30 min. The measurement was conducted by BD Accuri C6 flow cytometer. 3 × 10^4^ cells were analyzed for each sample.

### 4.7. ROS Production Assay

To evaluate the production of reactive oxygen species (ROS) in cells, after 96 h of co-incubation with OCP and OCP-Sr, the cells were washed three times with PBS solution, then detached from the surface of the culture plastic using 0.05% trypsin-EDTA solution and stained with 20 μM 2′,7′-dichlorodihydrofluorescein diacetate (DCFH-DA), (Sigma-Aldrich, St. Louis, MO, USA) for 15 min in a CO_2_ incubator [50]. As a control, cells were incubated with 1 mM hydrogen peroxide (Sigma-Aldrich, St. Louis, MO, USA) for 20 min. Fluorescence measurements were analyzed using a BD Accuri C6 flow cytometer. 3 × 10^4^ cells were analyzed for each sample.

### 4.8. Confocal Microscopy

Micrographs were obtained by planting cells on cover glasses in an amount of 0.5 × 10^4^ cells/cm^2^, cultured for 24 h, then OCP and OCP-Sr were added to cells and cultured for another 96 h. After 96 h of cultivation with OCP and OCP-Sr, cells were washed three times with PBS solution and stained with 1 μg/mL Hoechst 33342 (Sigma-Aldrich, MO, USA), 75 nM Lysotracker Green, and 200 nM MitoTracker Red CMXRos (Thermo Fisher Scientific, Waltham, MA, USA) for 20 min at 37 °C. After staining, cells were washed with Hanks’ Balanced Salt Solution (HBSS) (Gibco, Waltham, MA, USA). Micrographs were obtained using a TCS SP5 confocal microscope (Leica, Wetzlar, Germany).

### 4.9. Statistical Analysis

Results are presented as mean ± standard deviation (M ± SD). Each of the in vitro experiments was carried out at least five times (n ≥ 5). The statistical significance of the difference was determined using one-way ANOVA followed by multiple Holm–Sidak comparisons, *p* < 0.05. The design of experiment and related statistics (ANOVA) were carried out using SigmaPlot™ 14.0 (Systat Software Inc., San Jose, CA, USA). Plots were created using SigmaPlot™ 14.0.

## 5. Conclusions

Strontium substituted OCP can be obtained in a wide range of substitutions up to 25–27 at.% low-temperature method. The results of in vitro biocompatibility studies show that substitution of strontium in samples with OCP-Sr_20 and higher significantly increases cells viability, compared to undoped OCP. This effect may be associated with the absence of the OCP-Sr impact on the content of acidic compartments, as well as with the lack or even an inhibitory effect on the ROS production. The results of the study indicate the effectiveness of using strontium in OCP for improving biocompatibility of OCP-based composite materials intended for bone repair.

## Figures and Tables

**Figure 1 ijms-22-12747-f001:**
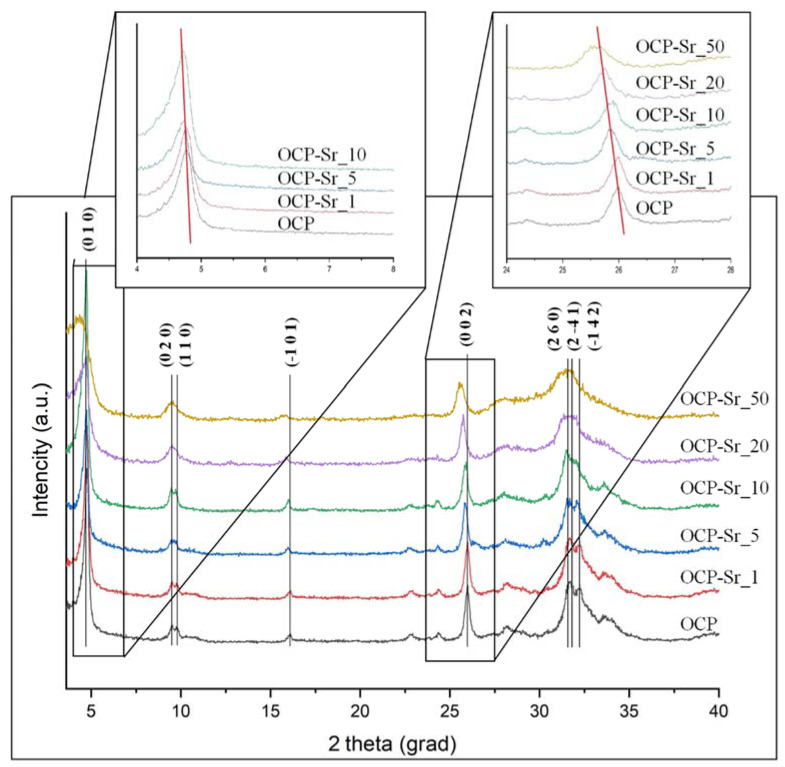
Powder X-ray diffraction patterns of the powder samples OCP and OCP-Sr.

**Figure 2 ijms-22-12747-f002:**
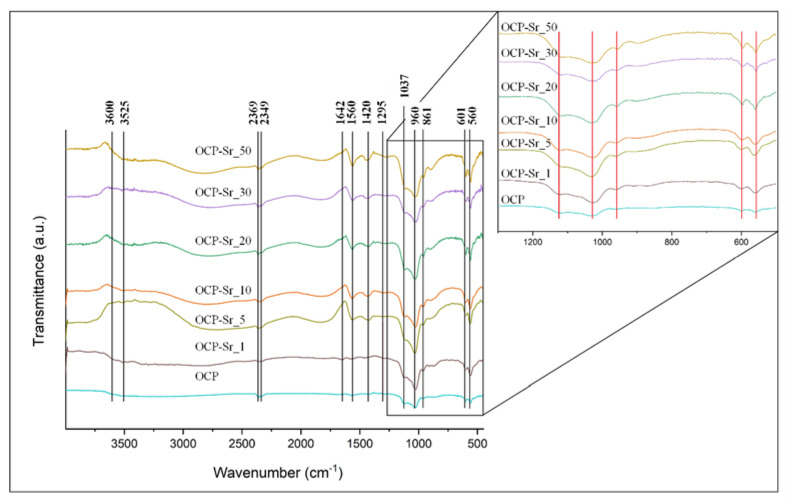
IR-spectra of samples OCP and Sr-OCP.

**Figure 3 ijms-22-12747-f003:**
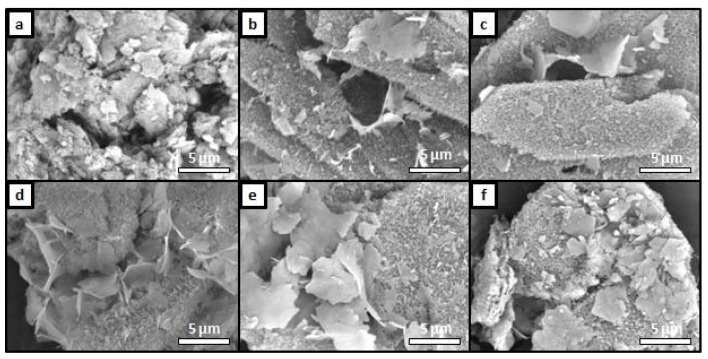
SEM micrographs of the OCP surface: (**a**) OCP; (**b**) OCP-Sr_1; (**c**) OCP-Sr_5; (**d**) OCP-Sr_10; (**e**) OCP-Sr_20; (**f**) OCP-Sr_50.

**Figure 4 ijms-22-12747-f004:**
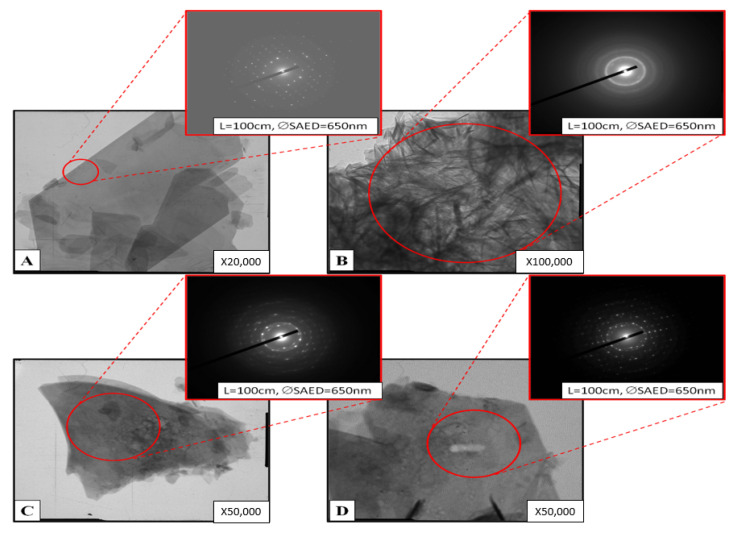
TEM images of characteristic samples (**A**,**B**) OCP and (**C**,**D**) OCP-Sr _10.

**Figure 5 ijms-22-12747-f005:**
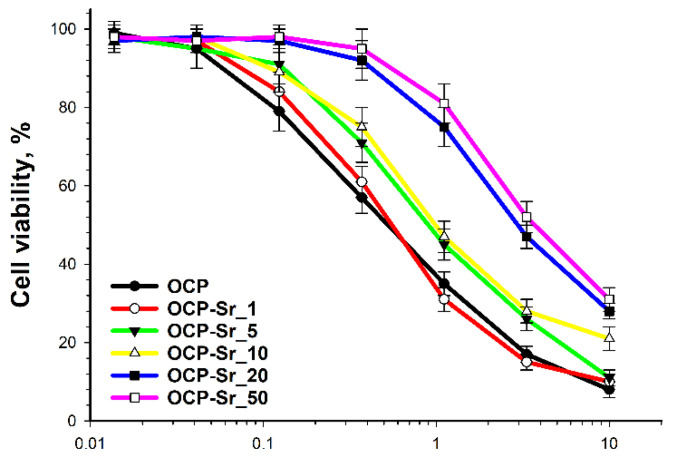
Viability of C3H/10T1/2 cells after 96 h of co-incubation with native OCP without strontium substitution for calcium (OCP) and Sr-substituted OCP (OCP-Sr) with varying degrees of substitution.

**Figure 6 ijms-22-12747-f006:**
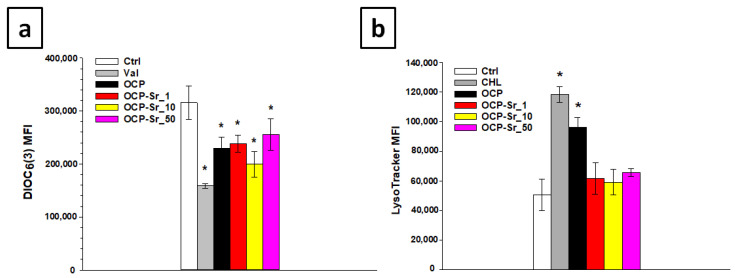
Mitochondrial membrane potential (**a**), quantitative assessment of the fluorescence intensity of acid compartments (**b**). Ctrl—untreated cells without the addition of calcium phosphate compounds. Val is valinomycin. CHL—chloroquine. MFI—Mean fluoresce intensity. *—statistically significant difference compared to untreated control cells, *p* < 0.05.

**Figure 7 ijms-22-12747-f007:**
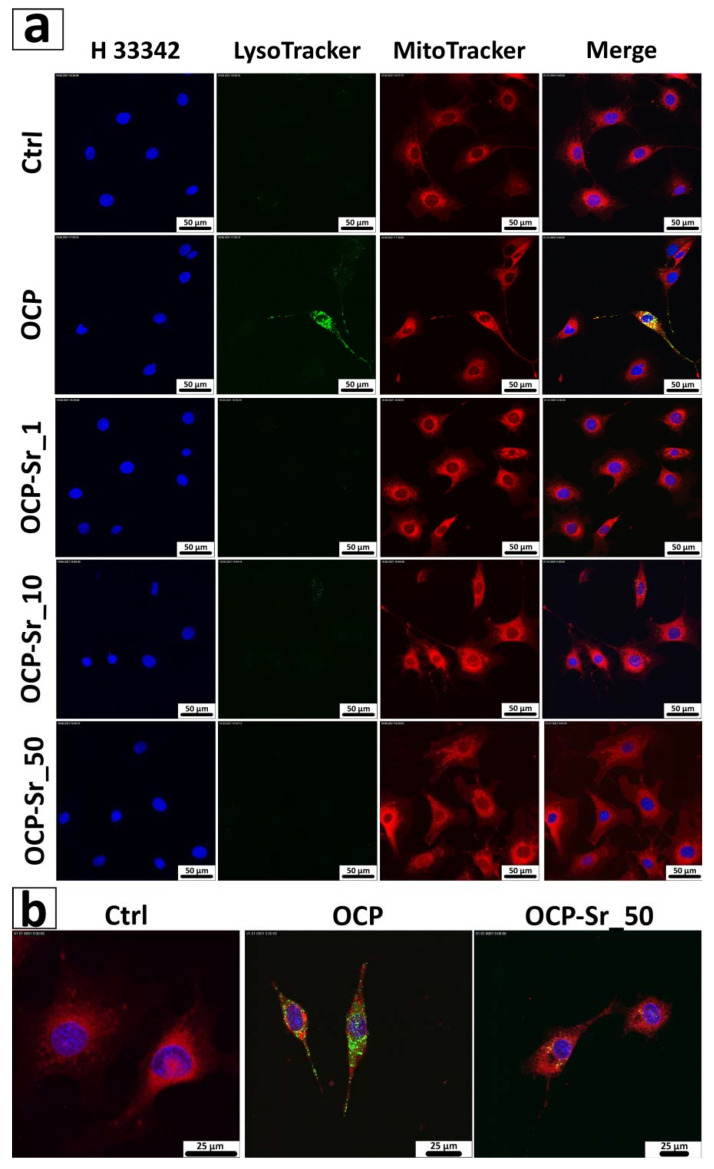
Confocal micrographs (**a**,**b**) of C_3_H/10T1/2 cells after 96 h of incubation with 1 mg/mL OCP and OCP-Sr. Cell nuclei are stained with Hoechst 33342 (blue), acidic compartments are stained with LysoTracker Green DND-26 (green), and mitochondria are stained with MitoTracker Red CMXRos (red).

**Figure 8 ijms-22-12747-f008:**
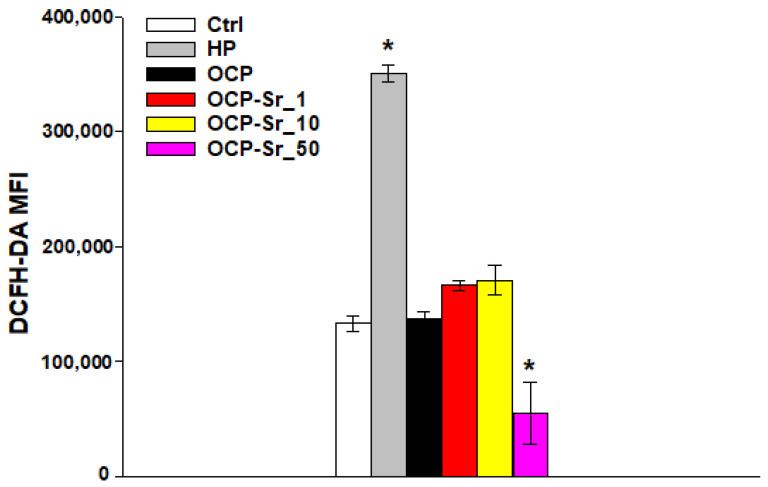
Production of reactive oxygen species (ROS) in C_3_H/10T1/2 cells after 96 h of incubation with OCP and OCP-Sr. Ctrl—untreated cells without the addition of calcium phosphate compounds. HP—hydrogen peroxide. MFI—Mean fluoresce intensity. *—statistically significant difference, compared with control cells, *p* < 0.05.

**Table 1 ijms-22-12747-t001:** Elementary cells parameters values. Standard deviations in parentheses.

Sample	Card № 26-1056	OCP	OCP–Sr_1	OCP–Sr_5	OCP–Sr_10
a, Å	9.52(9)	9.44(1)	9.62(2)	9.62(3)	9.62(7)
b, Å	18.99(4)	18.99(2)	19.01(1)	19.01(7)	19.02(4)
c, Å	6.85(5)	6.84(6)	6.89(4)	6.92(3)	6.90(3)
α, °	92.33(0)	92.52(1)	93.11(5)	93.17(4)	92.19(2)
β, °	90.13(0)	90.15(1)	89.20(7)	89.19(6)	90.18(3)
γ, °	79.93(0)	80.01(4)	80.09(3)	80.08(2)	79.78(9)
V, Å^3^	1220.5(7)	1220.5(6)	1240.2(6)	1240.2(2)	1238.2(2)

**Table 2 ijms-22-12747-t002:** Chemical components of samples OCP and OCP-Sr, at.%.

Sample	EDS					XRF				
	P	Ca	Sr	Ca/P	Ca + Sr/P	P	Ca	Sr	Ca/P	Ca + Sr/P
OCP-Sr_1	41.89	56.85	1.26	1.36	1.39	34.08	65.33	0.60	1.92	1.93
OCP-Sr_5	41.02	55.69	3.28	1.36	1.44	33.90	63.44	2.67	1.87	1.95
OCP-Sr_10	41.50	50.33	8.17	1.21	1.41	35.75	58.86	5.38	1.65	1.80
OCP-Sr_20	41.69	46.15	12.17	1.11	1.40	36.89	55.64	7.46	1.51	1.71
OCP-Sr_50	45.01	38.23	16.76	0.85	1.22	35.25	52.33	12.42	1.48	1.84

## Data Availability

Not applicable.

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
