# Peer review of "Octacalcium Phosphate for Bone Tissue Engineering: Synthesis, Modification, and In Vitro Biocompatibility Assessment"

_ijms, 2021, doi:10.3390/ijms222312747_

Round 1
Reviewer 1 Report
I think that the topic is of interest and worth to be published, but I think that there is one serious limitation. The discussion part is by too superficial; the limitations of the study are not really mentioned and discussed in a profound way. What is missing, what are the next steps,..? What is the real potential clinical impact,..? Please perform a more profound discussion!!
Author Response
Dear Reviewer
Thank you for your attention to our article, your comments helped us make it much better. Discussions of the potential application of the developed materials, as well as possible steps for further research are included in the article in the "Discussion" section.
--
Best regard,
Authors
Reviewer 2 Report
The following manuscript focus on OCP modification with Sr and its influence on physico-chemical and biological properties. In general the manuscript is well written and easy to understand, however some major corrections should be included to improve its quality. I would suggest to exted the discussion section as for now it is rather scarce and only partially allow to compare the obtained results with current state of the art in the field. I would also suggest to elaborate how the developed materials could be used in real life situation? What are the possible processing methods?
Minor comments:
Line 83 - Please explain sample labeling a bit better, as it is difficult to interpret labels used further in results section. The relation between Sr content or initial Sr addition and sample label is not clear.
Line 97 - I would suggest to change the order of equippment, as the current one suggests that sputtering machine is equipped with EDS, while I believe it should refer to SEM
Line 117 - Please explain how OCP and Sr-OCP was added to the cells - was that a previously prepared extract or the cells were incubated in the presence of samples? What were the concentrations used? How were the samples sterylized prior to addition to the cells? What was the excitation wavelength used in fluorescence measurements? How was cell viability calulated after the fluorescence intensity measurements?
Line 125 - Cell viability test was performed at a wide range of concentrations as I assume from the results, but furhter tests involving lysosome content etc. were clearly done for just one concentration. Please specify what was that concetration and why it was selected for those experiments.
Line 151 - Please uniform the number format (cell density in particular) across the manuscript.
Almost all figures - If possible, please improve figure quality as the labels are not clearly visible now.
Table 1 lacks the reference and expalantion in text.
Line 226 - please correct sentence grammar.
Figure 4 - a, b, c and d labels are a bit difficult to find, please make them similar to other figures.
Figure 6 - Please specify what is the unit at y axis as it is not clear enough - is it mean fluorescence intensty?. Fig. 6B - figure caption suggest that the presented values represent the number of lysosomes - how was that calculated?
Author Response
Dear Reviewer,
Please see the attachment.
--
Best regard,
Authors

Reviewer 3 Report
Comments to Authors:
In this study, the authors examined the effects Sr content in octacalcium phosphate (OCP) on the cell viability in vitro. The authors prepare OCP and Sr-containing OCP through the low temperature process and characterized the structure of these materials. Furthermore, the results of this study suggested that the incorporation of Sr in OCP lattice could improve the viability of mesenchymal stem cells in vitro. The examinations of the relationship between the material characteristics of Sr-containing OCP and cellar response are important to design the bone substitute materials. However, the completeness of manuscript is insufficient, and the manuscript cannot be accepted before several issues are addressed in my opinion.
1) Is it collect the angular range from 2θ = 8.0 to 54.4 degree in the analysis using XRD? Figure 1 displays XRD patterns of specimens in the range from 2θ = 4 to 40 degree.
2) The author described “The determination of the samples chemical composition was carried out by inductively coupled plasma (ICP) in an ULTIMA 2 instrument (France).” However, results in chemical composition determined using ICP were not shown in the manuscript.
3) The lattice parameters of OCP-Sr containing different amount of Sr2+ were estimated in this study. Improvement of the precision of calculated parameters displayed in Table 1 could be required to support the discussion “The findings indicate that the parameters of the crystal lattice gradually increase with an increase in the strontium concentration.”
The author should display the lattice parameters of OCP prepared under the low temperature in Table 1.
4) The results of XRD in this study suggested that increasing the content of Sr induced the lower crystallinity and lattice expansions. Could authors explain the details of “This may indicate that increasing strontium concentration stabilizes OCP structure.” in line 208?
5) The crystal stricture of specimens was analyzed by the selected area electron diffraction (SAED) (Figure 4). The SAED patters were obtained from OCP particles exhibiting two types of morphologies (Fig. 4a, b). In the SEM image (Figure 3d), OCP-Sr_10 also seemed to include the different shape of particles. Why did the author observe and analysis only the plate-like particles of OCP_Sr-10?
I think that the authors should also analyze the SAED patterns to determine whether crystal structure of each particle observed by TEM was OCP or not.
6) The results of cell culture experiments suggested that the increase in the content of Sr in the specimens associated with the decrease in number of lysosomes and inhabitation of ROS production in MSCs and to improve the cell viability. On the other hand, the size and morphology of particles as well as crystallinity of OCP were changed by the content of Sr (Figure 1, and 3). Could these factors regulate the production of lysosomes and ROS in vitro?
7) If Sr2+ ions contribute to the inhibition of ROS production and lysosome number, does the addition of free Sr2+ alone or with pure OCP improve cell viability? The author should discuss the mechanism of inhibition of ROS production and lysosome number by Sr2+, even if it is based on previous literature.
8) Previous literatures, such as Shi et al Bioactive Materials 6:1267-1282 (2021), have been examined the cellar response including the cell viability to Sr-doped OCP. The authors focused on the OCP prepared through the low temperature process using DCPD. Please discuss that the new findings of this study in terms of the characteristics of OCP and Sr-containing OCP prepared by the low temperature process for the cell viability compared to the OCP prepared by higher temperature process.
9) The symbol of the lattice parameter of “β” are not displayed correctly in Table 1. Furthermore, “.” should be used to indicate the decimal point in Table 1 and 2. Please clearly show the scale bars in the photographs of Figure 4 and 6.
10) The content of the text does not seem to match the references cited, for example “These data are well consistent with the results of E. Boanini, M. Gazzano, A. Bigi (2010) [23]…”. Please check the entire manuscript about the citations.
Author Response
Dear Reviewer
Please see the attachment.
--
Best regard,
Authors

Reviewer 4 Report
The manuscript “Octacalcium Phosphate for Bone Tissue Engineering: Synthesis, Modification and in Vitro Biocompatibility Assessment” presents physico-chemical and biological results on Sr-substituted OCP.
Some points need to be addressed to improve the manuscript:
1- I suggest to insert strontium-OCP in the title.
2- line 70- what is 50-100 microm?
3- lines 73-75- please indicate solid/liquid ratio
4- lines 76-77 and 81-82- please balance these chemical reactions properly
5- lines 78-79- please indicate Sr concentration
6- lines 73-84- please add experimental details: temperature, stirring rate, filtration/centrifugation…
7- line 180- what do the Authors mean with Sr_5/10/20? Where do these labels come from? Maybe linked to Sr concentration in solution?
8- line 181- the reference 23 is not correct, and the cited paper has been confused with DOI: 10.1021/cg100494f by most of the same Authors
9- table 2 – quantification of elements is presented here by EDS and XRF; whereas in Materials and Methods section ICP technique is described. Please revise.
10- table 2- What are these numbers? Atoms? Masses? Please, for each sample show data as: 1) Ca/P molar ratio; and 2) Sr%atoms over total cations (Ca+Sr). This would be more clear to the reader and would allow to compare data with those reported in the existing literature.
11- lines 285-294- usually evaluation of ROS reduction takes into consideration the amount of material as a variable. Please cite a reference for the method used in the present work.
Author Response

(The authors gave the same response as above.)

Round 2
Reviewer 2 Report
I am satisfied with the reviewed version of the manuscript.
Reviewer 3 Report
The authors have carefully addressed my concerns. I think that the revised manuscript is accepted for publication in International Journal of Molecular Sciences.